# Fault Diagnosis in the Slip–Frequency Plane of Induction Machines Working in Time-Varying Conditions

**DOI:** 10.3390/s20123398

**Published:** 2020-06-16

**Authors:** Ruben Puche-Panadero, Javier Martinez-Roman, Angel Sapena-Bano, Jordi Burriel-Valencia, Martin Riera-Guasp

**Affiliations:** Institute for Energy Engineering, Universitat Politècnica de València, Cmno. de Vera s/n, 46022 Valencia, Spain; jmroman@die.upv.es (J.M.-R.); asapena@die.upv.es (A.S.-B.); jorburva@die.upv.es (J.B.-V.); mriera@die.upv.es (M.R.-G.)

**Keywords:** analytic signal, wavelet transform, fault diagnosis, induction machines, analytic signal, spectrogram, time-frequency domain, Wigner–Ville distribution

## Abstract

Motor current signature analysis (MCSA) is a fault diagnosis method for induction machines (IMs) that has attracted wide industrial interest in recent years. It is based on the detection of the characteristic fault signatures that arise in the current spectrum of a faulty induction machine. Unfortunately, the MCSA method in its basic formulation can only be applied in steady state functioning. Nevertheless, every day increases the importance of inductions machines in applications such as wind generation, electric vehicles, or automated processes in which the machine works most of time under transient conditions. For these cases, new diagnostic methodologies have been proposed, based on the use of advanced time-frequency transforms—as, for example, the continuous wavelet transform, the Wigner Ville distribution, or the analytic function based on the Hilbert transform—which enables to track the fault components evolution along time. All these transforms have high computational costs and, furthermore, generate as results complex spectrograms, which require to be interpreted for qualified technical staff. This paper introduces a new methodology for the diagnosis of faults of IM working in transient conditions, which, unlike the methods developed up to today, analyzes the current signal in the slip-instantaneous frequency plane (s-IF), instead of the time-frequency (t-f) plane. It is shown that, in the s-IF plane, the fault components follow patterns that that are simple and unique for each type of fault, and thus does not depend on the way in which load and speed vary during the transient functioning; this characteristic makes the diagnostic task easier and more reliable. This work introduces a general scheme for the IMs diagnostic under transient conditions, through the analysis of the stator current in the s-IF plane. Another contribution of this paper is the introduction of the specific s-IF patterns associated with three different types of faults (rotor asymmetry fault, mixed eccentricity fault, and single-point bearing defects) that are theoretically justified and experimentally tested. As the calculation of the IF of the fault component is a key issue of the proposed diagnostic method, this paper also includes a comparative analysis of three different mathematical tools for calculating the IF, which are compared not only theoretically but also experimentally, comparing their performance when are applied to the tested diagnostic signals.

## 1. Introduction

Induction machines drive modern industry, and their failures, although very rare due to their robustness, may produce high economic losses because of the sudden breakdown of production lines. The continuous monitoring of their condition may help reduce this risk, by allowing an early detection of incipient machine faults. One of the main difficulties for implementing reliable fault diagnostic systems for induction machines is their variable speeds. Being asynchronous machines, their speed depends on the load conditions. In addition, variable speed drives are used to adapt their speed to production requirements. Speed variations imply the use of diagnostic techniques that are designed to operate in the join time-frequency domain, such as those presented in [1,2,3,4,5,6,7,8,9], based on the analysis of the machine current under transient conditions. These methodologies can be considered as an extension to the transient regime of the MCSA methodology, developed for the steady state regime of induction machines [10].

MCSA relies on the detection of the specific fault harmonics that each type of fault induces in the stator current, which may be considered their characteristic signature. In steady state, this signature can be detected in the spectrum of the stator current, obtained using the FFT [11,12,13,14,15]. Some of the most common induction machine faults and their characteristic signatures are:Rotor asymmetries, which generate fault related harmonics at frequencies given by [16,17]:
(1)$$f_{b}\left(s\right)=\left(1+2ks\right)f_{1}\;\;\;\;\;\mathrm{with}\;k=\pm 1,\pm 2,\ldots$$
where *s* is the slip, $s=\left(n_{s}-n\right)/n_{s}$, $n_{s}$ being the synchronous speed of the machine and *n* its actual speed), and $f_{1}$ is the power supply frequency.Mixed eccentricity, which generates fault harmonics at frequencies given by [18,19,20]
(2)$$f_{ME}\left(s\right)=f_{1}\pm \left(1-s\right)f_{1}/p=f_{1}\pm f_{r}$$
where $f_{r}$ is the rotational frequency of the motor, and *p* is the pole pairs number.Cyclic faults in the outer or inner races of bearings, which generate fault harmonics at frequencies that depend on the fault localization (inner or outer race), and on the bearing characteristics (number of balls, ball diameter, bearing pitch diameter, and the contact angle) [21,22], given by
(3)$$f_{B,o}=\left|f_{1}\pm \left(m\;\;0.4\;N_{b}\left(1-s\right)\;f_{1}/p\right)\right|=\left|f_{1}\pm m\;\;0.4\;N_{b}\;f_{r}\right|\;m=1,2,\ldots$$
(4)$$f_{B,i}=\left|f_{1}\pm \left(m\;\;0.6\;N_{b}\left(1-s\right)\;f_{1}/p\right)\right|=\left|f_{1}\pm m\;\;0.6\;N_{b}\;f_{r}\right|\;m=1,2,\ldots$$
for the outer and inner race, respectively, where $N_{b}$ is the number of bearing balls.

In MCSA, the current spectrum is commonly obtained using the fast Fourier transform (FFT), applied to the sampled stator current. In addition, the rotor speed must be accurately measured. The diagnostic process consists of detecting high amplitude harmonics at fixed frequencies, calculated with Equations (Equation 1), (Equation 2), (Equation 3), and (Equation 4). These Equations share a distinctive characteristic: they depend on the machine speed. Therefore, in transient conditions, with non-constant speed, the positions of the fault harmonics are no longer constant, which blurs their characteristic signatures (smearing effect) and hinders the application of steady state fault diagnostic techniques. In fact, as the frequency of these harmonics change with time, what Equations (Equation 1), (Equation 2), (Equation 3), and (Equation 4) provide is their instantaneous frequency (IF) [23,24].

This change from the single frequency domain to the joint time-frequency domain generates two main problems, whose solutions are proposed in this work:The choice of the tool for calculating the IF of the fault harmonics in variable speed conditions. It is needed to replace the FFT, used in steady state regime, with specialized transient regime techniques (time-frequency transforms, analytic signals, etc.), which are more complex and computationally intensive. Therefore, it is necessary to select a method with a good trade-off between computational burden and accuracy.The definition of suitable fault signatures in the time-frequency domain. It is needed to replace the analysis of fault harmonics amplitudes at fixed frequencies (steady state) with the analysis of their complex 2D trajectories in the time-frequency domain. Therefore, it is necessary to define time-frequency fault signatures with a good trade-off between complexity and sensitivity.

Regarding the first point, specialized time-frequency transforms that are able to display the fault harmonics evolution in the time-frequency domain, and generate their time-frequency spectra, have been proposed in the technical literature [25]: linear transforms, as the short time Fourier transform (STFT) [26,27], the wavelet transform (WT), both discrete (DWT) [28] and continuous (CWT) [29,30,31], the polynomial chirplet transform [32], the fractional Fourier transform (FrFT) [33], or the synchrosqueezing transform [27,34], and quadratic ones, as the the Wigner–Ville distribution (WVD) [35]. A different approach for tracking the time-evolution of the fault harmonics rely on the calculation of their IF using the Hilbert transform (HT) and the analytic signal [36,37,38,39,40,41,42,43,44]. It has been applied to the detection of stator turn faults [45], bearing faults [46], broken bars [23,47,48,49,50], planetary gearbox faults [51], or mechanical load faults [52].

Regarding the second point, new fault signatures are needed to perform fault diagnosis in transient regime. Direct evaluation of the amplitude and trajectory of fault harmonics in a 2D time-frequency plane is a challenging task, even for experienced maintenance personnel, much more difficult than assessing their amplitudes at specific frequencies as in steady state.

This work addresses both problems, and proposes a new methodology for the diagnosis of three types of usual faults in induction machines, which is based on tracking the IF of the fault related components of the stator current during the functioning of IMs in transient regime. Schematically, this methodology involves the following steps, summarized in the block diagram of Figure 1:Capturing one of the IM line currents and the rotor speed, sampling them with a suitable frequency $f_{s}$.Extracting the fault component related to the type of fault under study. The extraction of the fault component from the tested current is unavoidable, since the concept of IF is meaningless if one intent to apply it to complex multicomponent signals. The extraction process basically consists on filtering the tested current, keeping the part of the tested signal included in a frequency band that contains the range of frequencies that can be spanned by the fault component during the transient, and where it is predominant when the fault arises. There are different methods for the extraction of the fault components that basically consist on filtering techniques. In this work, the DWT is used for this purpose, since it enables in a simple an efficient way to take out frequency bands suitable for isolating the fault components.Calculating the IF of the subsignal extracted previously using a suitable mathematical tool.Obtaining the theoretical pattern *s*-IF corresponding to the fault component under study.Comparing the experimental *s*-IF plot with the corresponding theoretical pattern and formulating a diagnostic assessment. This comparison can be carried out in qualitative manner, but it is advisable to define a numerical parameter for fault severity assessment.

In this diagnostic scheme, steps iii and iv are critical issues, and require to be commented more deeply:Regarding step iii, as the proposed signatures are based on calculating the IF of the fault components along a transient functioning, a contribution of this paper is a thorough study comparing the main families of techniques for calculating the IF (linear time-frequency transform, quadratic time-frequency transform and the analytical function method). Three different techniques for extracting the IF are compared from a theoretical point of view, and also in a practical way, applying them to experimental tested currents of IM with different kinds of faults. This analysis allows conclusions to be drawn regarding the computational burden, the calculation time, and the precision of these techniques.Regarding step iv, the main contribution of this paper is to introduce new fault related signatures designed for transient functioning. Unlike the signatures currently used by diagnosis methods based on the above mentioned time-frequency approaches, the signatures here introduced are unique for every kind of fault. This means that they are the same for any induction machine, and do not depend on the way in which the load and speed vary. In fact, the fault signatures introduced in this paper are very simple, since they are straight lines, obtained by representing the IF of the fault harmonics given by Equations (1), (2), (3), and (4) not in the time-frequency domain, but in the slip–frequency domain; the slope of these straight lines is equal to the coefficient of the slip in Equations (1), (2), (3), and (4). This slope is different for each type of fault, and, therefore, Equations (1), (2), (3), and (4), understood as functions of slip, constitute suitable signatures not depending on IM or transient characteristics. The use of the proposed signatures significantly simplifies the diagnostic task, which can be carried out by non-specialized staff, and make easier the application of automatic diagnostic algorithms based on IA techniques. These signatures are theoretically justified and experimentally tested.

This paper is structured as follows: In Section 2, three different methods for calculating the IF of a signal component (the continuous wavelet transform, the Wigner–Ville transform, and the analytic signal), are presented, both theoretically and in a practical way, applying them to a simulated machined with a broken bar fault. In Section 3, the three methods for calculating the IF are experimentally tested on commercial machines, with different types of forced faults: on an induction machine with a broken bar, on a machine with mixed eccentricity and on a machine with a cyclic bearing defect. In Section 4, the new signatures proposed are explained, and they are applied to the diagnosis of the faults of the machines tested in Section 3. In addition, they are used to compare the three methods of calculating the instantaneous frequency, to choose the best one. In Section 5, a series of practical remarks related to the application of the method are commented. Finally, Section 6 presents the conclusions of this work.

## 2. Presentation of the Methods for Obtaining the IF of the Fault Components in the Slip–Frequency Domain

Among the many different approaches for calculating the IF of time-varying signal components [53], three different methods will be compared in this work:The derivative of the phase of the analytic signal (AS) of the stator current component.The first conditional moment of frequency for a given time, obtained from quadratic time-frequency energy distributions, such as the Wigner–Ville distribution (WVD).The evaluation of the ridges in linear time-frequency transforms (CWT with the Morlet Wavelet).

These three methods will be presented briefly both theoretically and practically, using one of the principal fault harmonics of a simulated induction machine with a broken bar. This synthetic signal is the fault harmonic corresponding to a value of k=−1 in Equation (Equation 1), which is commonly known as the low side fault harmonic (LSH). The transient condition simulated has been the start-up transient of the motor. The start-up current is shown in Figure 2 (top). During this transient, the slip of the machine varies in the range s=1 to s≈0, and generates a characteristic V-shaped LSH start-up harmonic (LSHst), which has been represented in Figure 2 (bottom).

The process for generating this test signal has been the following one:First, the start-up of a 4-pole, 4 kW cage motor with a broken bar is simulated using the commercial finite-elements software Flux-2D. Figure 2 (top) shows the simulated current. More details about the characteristics of the induction motor are given in Appendix A.Second, the LSHst is isolated from the current signal using the DWT [54]. More accurately, the signal of Figure 2 (bottom) is the approximation of level 4 (a4) of the wavelet decomposition of the current of Figure 2 (top), using Daubechies-44 as mother wavelet. Since the time increment used in the simulation is dt = 0.001 s, the sub-signal a4 is formed by the components of the current included in the interval [0, 31.25] Hz; since in a machine with broken bars, the LSHst is by far the more prominent component in the frequency band below the supply frequency ($f_{1}$ = 50 Hz) [54], Figure 2 (bottom), can be taken as a good representation of the LSHst in that interval.

The procedure for presenting the three selected methods consists of applying each method to obtain the IF of the LSHst of Figure 2 (bottom) in the slip–frequency plane, and comparing it with the theoretical IF given by Equation (Equation 1).

### 2.1. Theoretical IF of the LSHst in the Slip–Frequency Plane

The theoretical IF of the LSHst in the slip–frequency plane, applying Equation (Equation 1) during the start-up transient, consists only in two straight segments, with a slope equal to $-2f_{1}$ (which becomes positive for s>0.5 because the "negative" frequencies given by Equation (Equation 1) from this point on, appear physically as positive ones ). This theoretical pattern has been plotted in the red line in the figures corresponding to each of the analyzed methods.

### 2.2. IF of the LSHst Using the Hilbert Transform of the Current

The calculation of the IF of the LSHst using the Hilbert transform is fully presented in [23]. Briefly, it consists of the following steps:Application of the Hilbert transform to the LSHst fault harmonic, a current signal designed as $i_{Ls}\left(t\right)$:
(5)$$\mathrm{HT}\left(i_{Ls}\left(t\right)\right)=\frac{1}{\pi t}i_{Ls}\left(t\right)=\frac{1}{\pi }\int_{-\infty }^{+\infty }\frac{i_{Ls}\left(\tau \right)}{t-\tau }\;d\tau$$Construction of the analytic signal of the LSHst, $i_{as}\left(t\right)$. It is a complex signal with a real part equal to the original signal $i_{Ls}\left(t\right)$, and an imaginary part equal to its HT:
(6)$$i_{as}\left(t\right)=i_{Ls}\left(t\right)+j\mathrm{HT}\left(i_{Ls}\left(t\right)\right)=A\left(t\right)e^{j\varphi (t)}$$Once obtained the complex AS, the derivative of its phase gives the IF of the LSHst:
(7)$$\mathrm{IF}\left(t\right)=\frac{1}{2\pi }\frac{d\varphi (t)}{dt}$$

The IF of the simulated LSHst of Figure 2 (bottom), obtained with this method, is shown in Figure 3. The dashed red lines represent the theoretical evolution of the IF. The calculated IF fits perfectly the theoretical one for frequencies below 31.25 Hz. The parts of the actual LSHst with frequencies above this value are not included in the signal of Figure 2 (bottom), due to the method used for extracting it. This is why there is no coincidence between the theoretical and the calculated IF for frequencies greater than 31.25 Hz.

### 2.3. IF of the LSHst Using the Wigner–Ville Distribution

The distribution of the energy of the LSHst in the joint time-frequency plane can be obtained applying a time-frequency distribution to this signal [55], and joining the points with the highest amplitude. The Wigner–Ville distribution, as other quadratic distributions, is particularly well suited for tracking signals with a linear variation of the IF, such as the LSHT. The time-frequency distributions presented in this paper have been calculated using the code given by [56]. The WVD belongs to a wider group of time frequency distributions [55], known as Cohen’s class distributions. One of the properties of these transforms is that the first conditional moment of frequency for a given time (the average of the frequencies existing in the time-frequency plane for a given time) gives exactly the IF of the signal, as defined by Equation (Equation 7). The WVD is the simplest transform of this group. It is given by:(8)$${\mathrm{WVD}}_{x}\left(t,f\right)=\int_{-\infty }^{+\infty }x^{*}\left(t-\frac{\tau }{2}\right)x\left(t+\frac{\tau }{2}\right)e^{-j2\pi f\tau }d\tau$$

The WVD has been applied to the simulated LSHst of Figure 2 (bottom), and the result is presented in Figure 4. A disadvantage of the WVD, and in general of the quadratic transforms, is the presence of interference cross-terms, which can be observed in Figure 4. More complex transforms of the Cohen’s class have been designed to minimize these cross-terms, as the smoothed pseudo Wigner distribution (SPWD) and the Choi–Williams distribution (CWD) [57]. Nevertheless, the cross-terms hardly affect the first conditional moment when calculating the IF, as shown in Figure 4, which justifies the choice of the WVD as the simplest one in the Cohen’s group for calculating the IF of the fault harmonics.

Figure 5 shows the IF that is obtained using the first conditional moment of frequency of the WVD, applied to the spectrogram of Figure 4. It has been plotted against the slip. As predicted theoretically, the IF can be recovered with this method, even under the presence of numerous cross-terms, which makes the visual interpretation of the image diffficult.

### 2.4. IF of the LSHst Using the Continuous Wavelet Transform

The distribution of the energy of the LSHst in the joint time-frequency plane can also be obtained using a linear time-frequency transform, which is free from cross-term artifacts. One of such distributions is the CWT [58], which is used to build the reassigned scalogram of the LSHst in this work.

The CWT, as defined by Equation (Equation 10), analyzes the LSHst using basis functions with compact support. They can be obtained through a process of dilatations and translations of a single base function, the mother wavelet ψ(t). In this work, the mother wavelet used is the Morlet wavelet, given by Equation (Equation 9)
(9)ψ(t)=11πfbπfbe−t2/fbej2πfct
where $f_{c}$ is the central frequency of the wavelet and $f_{b}$ is a bandwidth parameter (in this paper, $f_{c}$ = 1 Hz and $f_{b}$ = 2 Hz).

The energy distribution of the LSHst, its scalogram, is obtained as
(10)$${\mathrm{CWT}}_{x}\left(t,a\right)=\frac{1}{\sqrt{a}}\int_{-\infty }^{+\infty }x\left(\tau \right)\psi^{*}\left(\frac{t-\tau }{a}\right)d\tau$$
and it has been represented in Figure 6. The IF of the LSHst is calculated based on the ridges of the scalogram, the local maxima [59]. The resulting IF is shown in Figure 7.

## 3. Experimental Comparison of the Methods for Calculating the IF of the Fault Components in the Slip–Frequency Domain

The scheme of the experimental test rig used in this work is shown in Figure 8. Two commercial machines are tested, whose data are given in Appendix A and Appendix B. During each test, the machine is directly coupled to a direct current (DC) machine rated 10 kW; the stator current is sampled with a current clamp (see Appendix C) connected to the analog voltage input module (ref. 701250, 10 Ms, 12 bit) of a digital oscilloscope Yokogawa DL750. The speed is also recorded using a a 200 pulse/revolution encoder connected to a frequency input module (ref. 701280), which directly calculates and records the speed (measurement range 0.01–100,000 rpm). The acquired data have been processed using a computer whose characteristics are given in Appendix D.

The test rig shown in Figure 8 will be used for obtaining the IF of fault harmonics, using the three presented methods, in case of a machine with broken bars (Section 3.1), a machine with a mixed eccentricity fault (Section 3.2), and a machine with a cyclic bearing fault (Section 3.3).

### 3.1. IF of the LSHst Extracted from the Tested Startup Current of a Machine with Broken Bars

In this section, the three selected methods are applied to the calculation of the IF of the principal fault harmonic (LSHst) of the tested current of a commercial induction motor with a provoked broken bar fault. Figure 9 shows the stator line current signal acquired during a direct online start-up of the 4-pole induction motor (see Appendix B) with a provoked broken bar, sampled at a rate of $f_{s}$ = 5000 samples/s. Figure 10 shows the experimental LSHst harmonic during the start-up, which have been extracted from the current of Figure 9 using the DWT-based procedure described in [54].

Figure 11 (from top to bottom) shows the experimental IF obtained through the application of the three methods presented in Section 2 to the LSHst of Figure 10, and compares the results with the theoretical evolution of its IF in the slip–frequency plane.

The similitude of the three graphics in Figure 11 and the good concordance of all of them with the theoretical evolution of the IF (dashed red lines in Figure 11) show that the three techniques of calculating the IF are suitable for diagnostics purposes, and also that the proposed IF based method for the diagnosis of bar breakages does not depend of the mathematical tool used for calculating the IF, but on the physical background explained in [23].

For comparison purposes, the same analysis has been applied to a healthy machine, and the result is shown in Figure 12. In this case, the experimental IF of the LSHst extracted from the motor current, using the AS method, does not coincide at all with the theoretical evolution of the IF in a machine with broken bars. This indicates the absence of such type of fault in the tested healthy motor.

### 3.2. IF of the Fault Component Extracted from the Tested Startup Current of a Machine with Mixed Eccentricity

In this section, first, the theoretical evolution of the IF of the fault harmonics in a machine with mixed eccentricity will be explained, and, then, it will be experimentally evaluated by testing a commercial machine with a provoked eccentricity using the three methods under evaluation.

#### 3.2.1. Theoretical IF of the Mixed Eccentricity Fault Related Harmonics in the Slip–Frequency Plane

The theoretical IF of the fault harmonics generated by a ME fault in the slip–frequency plane, applying Equation (Equation 2), consists only of two straight segments, with a slope equal to $\pm f_{1}/p$. In the case of the motor used in this section (Appendix B), which is a two pole pairs machine, p=2, so Equation (Equation 2) becomes
(11)$$f_{ME}\left(s\right)=f_{1}\pm \left(1-s\right)f_{1}/2$$
and the theoretical slope of the fault harmonics is $\pm f_{1}/2$.

#### 3.2.2. Experimental IF of the Mixed Eccentricity Harmonics in the Slip–Frequency Plane

In this section, the three previously exposed methods for tracking the IF are applied to calculate the IF of the fault components produced by a mixed eccentricity fault during a start-up transient, using the commercial induction motor whose characteristics are given in Appendix B.

To provoke this type of fault, the original bearings of the motor (see Figure 13a) have been substituted by new ones (Figure 13d), with a smaller outer diameter and a greater inner diameter. Two precision eccentric machined steel rings (Figure 13e) and Figure 13c) have been used for adjusting the new bearing to the bearing housing (Figure 13e) and to the shaft (Figure 13c). In this way, the cylindrical surfaces of both rings become eccentric, 0.15 mm for the outer ring b, and 0.25 mm for the inner ring c. Figure 13e shows the assembly mounted on the shaft, which provokes a mixed eccentricity fault with 30% of static eccentricity and 50% of dynamic eccentricity.

The current and the speed recorded during the start-up transient are shown in Figure 14. For calculating the IF of the mixed eccentricity components, it is necessary to extract them from the start-up current signal; the extraction of these components is more complex than in the case of bar breakages because their amplitude is substantially smaller. In this paper, the extraction of the ME components during the start-up is performed following the process described in [60]: first, the Hilbert transform is applied to the current signal for obtaining its envelope, the instantaneous amplitude of the analytic signal (see Figure 15); then, the eccentricity components are easily extracted by applying a high pass filter to this envelope, which behaves as a quasi-continuous signal.

This procedure demodulates effectively the current signal, by converting the mains frequency into a DC component, which can be easily filtered out. The envelope signal also contains the same fault harmonics as the original current signal, but they appear with their characteristic absolute frequencies, instead of modulating the mains frequency, as derived analytically in [61]. Therefore, the current envelope effectively demodulates the current signal. When this method is applied to the extraction of the mixed eccentricity harmonics, the two fault components given by Equation (Equation 2), which in the start-up current evolves within the intervals [≈ 25, 50] and [50, ≈ 75] Hz, are merged into a single fault component included into the frequency band [0,25] Hz, with a characteristic frequency
(12)$$f_{ME,AS}\left(s\right)=\left(1-s\right)f_{1}/p=f_{r}$$
which has been represented in Figure 16. From Equation (Equation 12), the characteristic slope of this harmonic is $-f_{1}/p$, that is, the diagnostic process is simplified because only one fault harmonic must be tracked, instead of the two fault harmonics present in Equation (Equation 2).

The extraction of the eccentricity components from the envelope of the start-up current is performed by means of the DWT (Figure 17). Taking into account the sampling frequency used, $f_{s}$ = 5 kHz, a decomposition with 10 levels is suitable for this end; Daubechies-10 was used as mother wavelet. The eccentricity components are included into the signals d7, d8, d9, d10, and a10, covering the frequency band [0, 39.06] Hz. The signal used for calculating the IF is obtained as:(13)$$i_{exc}=d7+d8+d9+d10$$

The approximation signal a10 has not been included in Equation (Equation 13), since it contains the envelope component due to the variation of the fundamental component of the current, which would mask the envelope components related with the eccentricity.

The signal $i_{exc}$ which will be used for calculating the IF through the three different methods is shown in Figure 18 (top), along with its Wigner–Ville distribution, Figure 18 (bottom). The initial and final points of this signal have been eliminated, for avoiding the border effects of the DWT. The eccentricity component is easily identifiable in the spectrogram, although it is surrounded by cross terms; its frequency increases from 0 to ≈25 Hz during the start-up ( 0<t<4 s) and remains constant from this instant.

Figure 19 shows the evolution of the IF versus the slip of the eccentricity components of Figure 18, calculated by the HT (top), the WVD (middle), and the reassigned scalogram obtained via CWT (bottom).

In Figure 19, it is shown that the IF calculated with the three techniques follows clearly the trend predicted by the theoretical evolution (dashed straight lines in Figure 19).

### 3.3. IF of the Fault Component Extracted from the Tested Startup Current of a Machine with a Bearing Fault

In this section, first, the theoretical evolution of the IF of the fault harmonics in a machine with a bearing fault (a cyclic fault in the outer race) will be presented, and then it will be experimentally obtained testing a commercial machine with a provoked fault using the three methods under evaluation.

#### 3.3.1. Theoretical IF of the Bearing Fault Harmonics in the Slip–Frequency Plane

The theoretical IF of the main fault harmonics generated by a bearing race fault is a straight segment starting at the point (s=1, $\mathrm{IF}=f_{1}$), with a slope of $-m0.4N_{b}f_{1}/p$ for the outer race fault given by Equation (Equation 3), and $-m0.6N_{b}f_{1}/p$ for the inner race fault, given by Equation (Equation 4). Figure 20 shows the theoretical IF of the outer race fault (solid line), and the inner race fault (dashed line), for a machine with p=2 pole pairs, $N_{b}=9$ balls in the bearings, taking $f_{1}=50$ Hz and m=1.

#### 3.3.2. Experimental IF of the Bearing Fault Harmonics in the Slip–Frequency

For the experimental validation of the proposed diagnostic method, a hole in the outer race of one of the bearings (model SKF 6205) of the tested motor (Appendix B) has been artificially produced. The geometrical specifications of the bearing are: ball diameter $d_{b}$ = 7.94 mm, pitch diameter $D_{b}$ = 39.04 mm, number of balls $N_{b}$ = 9, and contact angle Φ = 0. The stator current has been sampled at a frequency of 3 kHz. As in the case of the eccentricity test, the motor’s current has been demodulated using the HT. When this method is applied to the extraction of the outer race fault components, both fault components corresponding to *m* = 1 in (3), which in the start-up current evolves following the straight trajectories in the slip–frequency plane 50→0→40 Hz and 50→140 Hz, are merged into a single fault component with a characteristic frequency [61]:(14)$$f_{B,o}\left(s\right)=0.4\cdot N_{b}\cdot \left(1-s\right)f_{1}/p=0.4\cdot 9\cdot \;f_{r}=3.6\cdot f_{r}$$

During a start-up transient with $f_{1}$ = 50 Hz, *p* = 2, $f_{B,o}$ is included into the interval [0,90] Hz. Figure 21 is the representation of Equation (Equation 14) in the slip–frequency plane during a start-up transient.

The extraction of the fault component has been performed by means of the DWT, as in Figure 17. Taking into account the sampling frequency $f_{s}$ = 3 kHz, a decomposition with 10 levels is suitable for this end; Daubechies-10 has been used as mother wavelet. The bearing fault components are included into the signals d5, d6, d7, d8, d9, and d10, covering the frequency band [2.93, 93.75] Hz. The signal used for calculating the IF is obtained as:(15)$$i_{outer_{r}ate}=d5+d6+d7+d8+d9+d10$$
and it is shown in Figure 22 (top), along with its Wigner–Ville distribution, Figure 22 (bottom).

Figure 23 shows the evolution of the IF versus the slip of the outer race fault components of Figure 22, calculated by the HT (top), the WVD (middle), and the reassigned scalogram obtained via CWT (bottom).

## 4. Definition of a Fault Signature Based on the IF of the Fault Components

In the previous section, three different mathematical tools for calculating the IF of the fault components have been compared, but in a qualitative way. Therefore, this Section first introduces an analytical parameter for evaluating in a quantitative way the fitting between the calculated IF graphs and the theoretical pattern. This parameter is then used to objectively compare the precision of the three IF calculation methodologies. After that, the methods for calculating the IF will be compared from the point of view of the computational costs, characterized by the required time of computation.

An objective comparison of the ability of each method for calculating the right value of the IF will be performed through the parameter $R_{\mathrm{IF}}$ (correlation parameter) defined by Equation (Equation 16). This parameter compares the slope of the straight line obtained from a linear regression fit of the experimental frequency-slip points, and the slope of the theoretical IF pattern associated with the fault component:(16)$$R_{IF}=1-|\frac{Experimental\;\mathrm{IF}\;Slope-Theoretical\;\mathrm{IF}\;Slope}{Theoretical\;\mathrm{IF}\;Slope}|$$

Table 1 compares the values obtained for the parameter $R_{\mathrm{IF}}$ obtained from the three methods used in this paper. The results obtained with the healthy machine have also been included at the bottom of Table 1 for comparison purposes.

The parameter $R_{\mathrm{IF}}$ can be used as a fault signature parameter, since for all types of fault and IF calculating methods, it reaches values close to 1 when the fault exists and remains close to 0 when the machine is in healthy condition; this parameter summarizes in a simple figure the behavior of the IF along all the transient analyzed and thus brings a high reliability against false positive diagnostics: it is unlikely that a random perturbation could lead to a value $R_{IF}\approx 1$ since this would mean that the evolution of his IF would fit the theoretical one during all the successive regimes covered by the tested transient. It must be pointed out that the IF, as it is used in this paper, does not inform about the severity of the fault; nevertheless, the algorithms used for calculating the IF also enables calculating the instantaneous amplitude of the fault components; thus, it would be possible to define parameters based on the energy of the fault components, similarly to that defined in [23] for evaluating the severity of the faults, but this task is not addressed in this paper.

About the computational cost, Table 2 summarizes the average time needed by the three methods analyzed in this section to compute IF of the fault harmonics, using the computer given in Appendix D. All the times are shown relative to the AS method.

The most time-efficient of the compared methods is the AS. On the other hand, as it can be observed comparing Figure 3, Figure 5, Figure 7, Figure 11, Figure 19, and Figure 23, the wavelet scalogram produces a smoother representation of the IF of the fault harmonics.

A remarkable observation is that the derivative of the phase of the AS and the first moment of frequency of the WVD provide nearly identical results, as can be seen in Figure 11, Figure 19 and Figure 23, in spite of having been obtained using totally different mathematical methods, and also in spite of the many cross-terms that affects the WVD time-frequency representation.

## 5. Practical Remarks

In this paper, the tests for the validation of the proposed approach have been performed using the direct on line start-up transient. This transient has been selected since it is an actual functioning condition undergone by many industrial machines and, furthermore, it enables the validation of the method in a wide range of slips. Nevertheless, the theoretical basis of the method is applicable under any transient that imposes a slip variation, such as plugging, soft-starter start-up, or even load variations; however, the application of the method to transients with a limited range of slip variation implies some practical problems which are under study. Figure 24 shows the IF of the LSH obtained from the tested machine, whose characteristics are given in Appendix B, with one broken bar, under plugging conditions; the IF also evolves in this case as a straight segment with slope equal to $2f_{1}$, as predicted by Equation (Equation 1), giving a computed correlation parameter $R_{IF}=0.967$.

Related to the suitable duration of the transient, there is a practical limitation for the reliable application of the method: the necessity of a minimum length of the transient used to apply it. All transient regimes in electrical machines start with an electromagnetic transient period, in which the currents of the phases are far from constituting a balanced three-phase system. Under these conditions, the concept of fault components is meaningless, and thus the instantaneous frequency lacks significance. Therefore, for applying the proposed approach, it is necessary that the length of the electromagnetic transient will be short enough with respect to the total duration of the transient regime used for the applying the method. In the case of start-up transient, and based on the authors’ experience, times around 0.2 ≈ 0.3 s for the electromagnetic transients are usual and thus times greater than 1 s could be adopted as reference for the application of the proposed approach. Nevertheless, this limitation is not very important, bearing in mind that the more interesting application field of the methods of diagnostic based on transient analysis is the high power machines, for which starting times of several seconds or tens of seconds are usual.

The proposed methodology is well suited for being used in predictive maintenance programs of important machines, usually combined with other diagnostic methodologies. The diagnosis approach as is introduced in this work can be applied every time that the machine undergoes a transient (usually a start-up), checking if the stresses produced during the transient have altered the machine condition. Based on the authors’ experience, the application of the approach based on the reassigned scalogram can take a time around several minutes. If the algorithm based on HT is used, the computation time is reduced to a fraction of second. The computational requirements in this case are not excessive and a digital signal processor (DSP) could be sufficient for implementing the proposed approach.

## 6. Conclusions

In this paper, a set of simple signatures for the characterization of fault harmonics associated with different kinds of faults in induction machines working in transient regime has been presented. One of the proposals of this work is the replacement of the time-frequency by the slip–frequency plane as the diagnostic domain. It has been theoretically explained and experimentally validated that the IF of the fault harmonics follows straight lines in this plane, with slopes that are specific for each type of fault. The other proposal presented in this paper is the definition of a suitable fault signature parameter, which reflects in a single figure the presence of a fault during the operation in transient regime. Additionally, in this work, three different methods for calculating the IF of the fault harmonics have been evaluated, both with simulated signals and with experimental currents: the derivative of the phase of the AS of the current fault components, the first conditional moment of frequency of the WVD, and the CWT scalogram. The three methods have been successfully used for measuring the IF of the LSHst extracted from the start-up current of a simulated and a tested cage motor with one broken bar, and for measuring the IF of the fault components of a tested eccentric motor and of a motor with a localized point fault in the outer race of one bearing. The AS method has been selected as the most efficient one.

The diagnosis of induction motors fed from variable speed drives, or wind generators operating under variable wind speeds, are promising fields for the application of the proposed method.

## Figures and Tables

**Figure 1 sensors-20-03398-f001:**
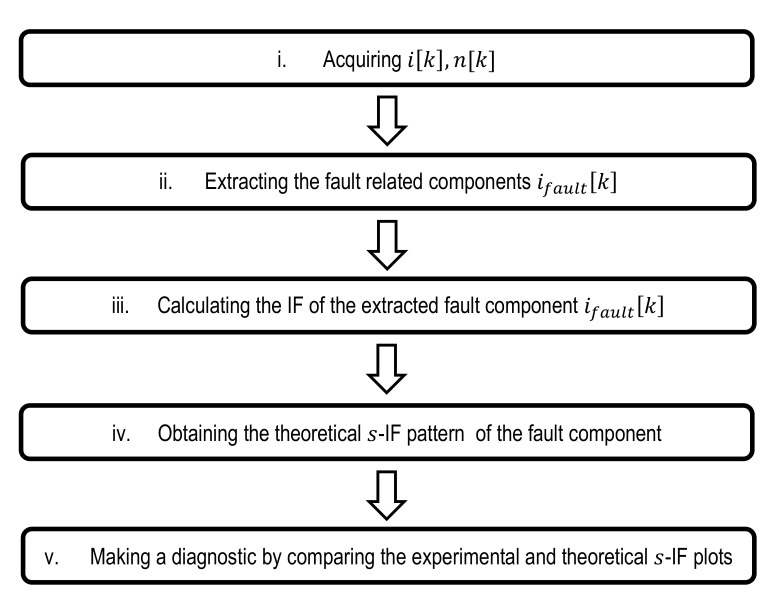
Block diagram of the diagnostic methodology proposed in this work.

**Figure 2 sensors-20-03398-f002:**
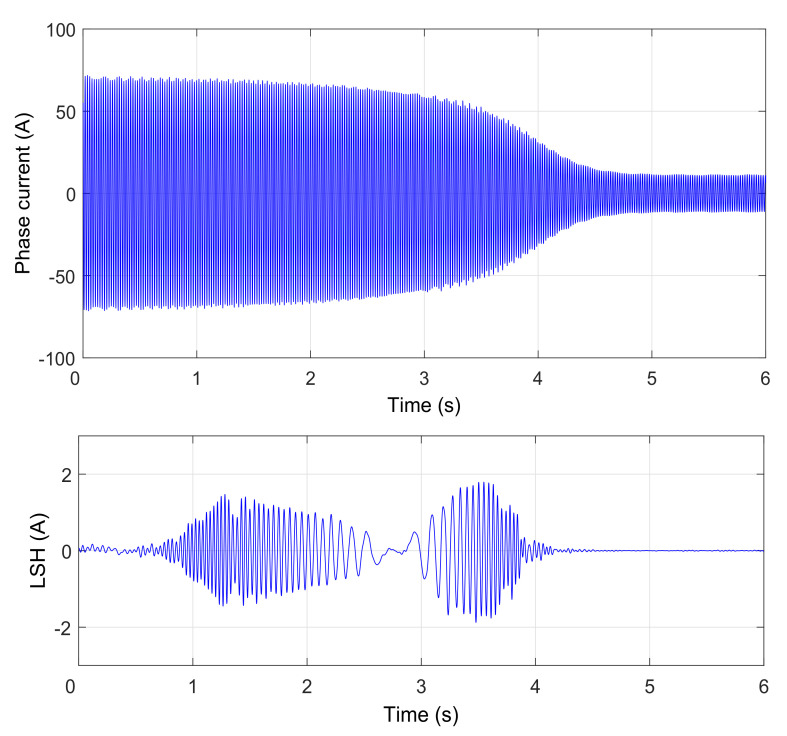
Simulated start-up current of an induction motor with one broken bar (**top**) and its corresponding LSHst (**bottom**).

**Figure 3 sensors-20-03398-f003:**
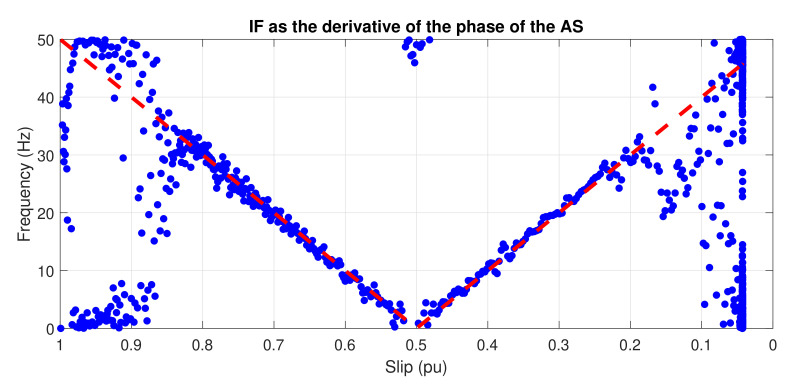
Instantaneous frequency of the LSHst versus slip, computed through the analytic signal. Dashed, red line: theoretical evolution of the fault harmonic. Blue dots: evolution of the LSHst of the simulated machine.

**Figure 4 sensors-20-03398-f004:**
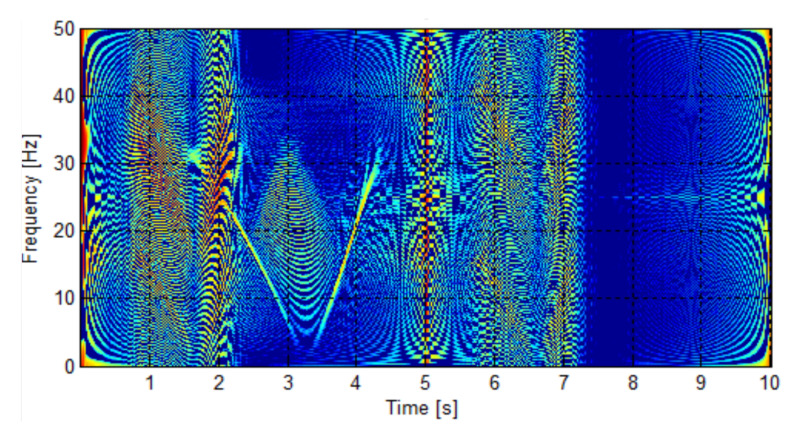
Wigner–Ville distribution of the LSHst of the simulated machine, in the time-frequency plane.

**Figure 5 sensors-20-03398-f005:**
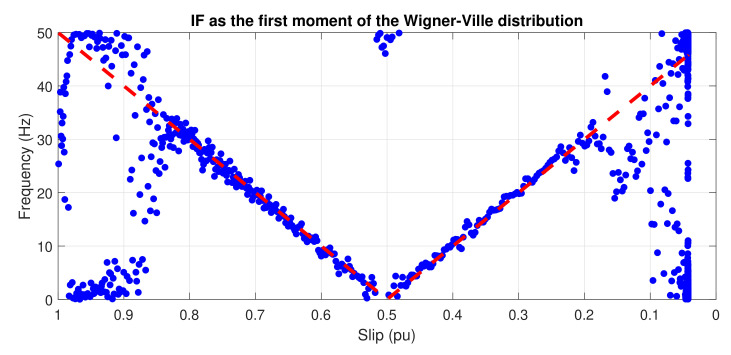
Instantaneous frequency of the LSHst in the slip–frequency plane, obtained from the WVD. Dashed, red line: theoretical evolution of the fault harmonic. Blue dots: evolution of the LSHst of the simulated machine.

**Figure 6 sensors-20-03398-f006:**
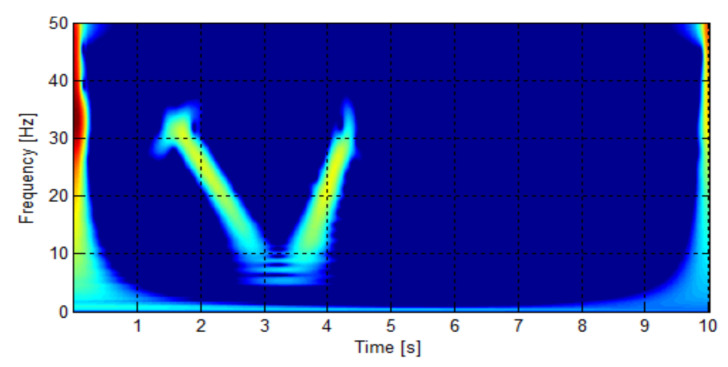
Reassigned scalogram of the LSHst in the simulated machine, in the time-frequency plane.

**Figure 7 sensors-20-03398-f007:**
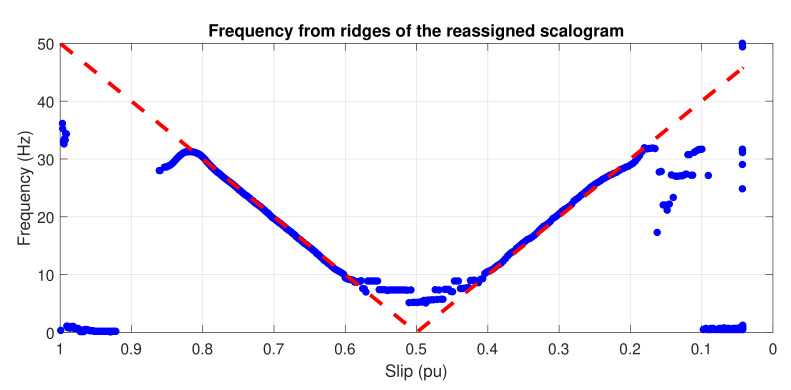
Instantaneous frequency of the LSHst in the slip–frequency plane, obtained from the scalogram ridges. Dashed, red line: theoretical evolution of the fault harmonic. Blue dots: evolution of the LSHst of the simulated machine.

**Figure 8 sensors-20-03398-f008:**
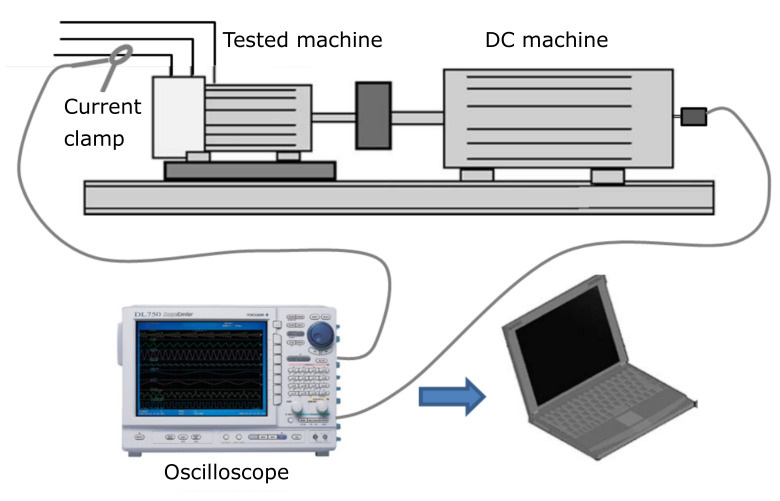
Schema of the test rig.

**Figure 9 sensors-20-03398-f009:**
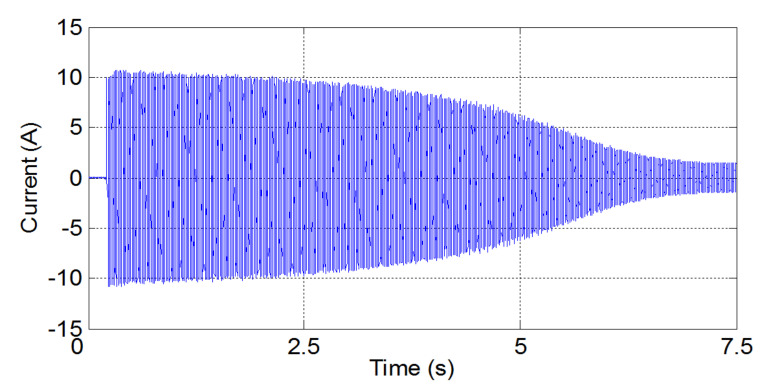
Tested start-up current of the machine with one broken bar.

**Figure 10 sensors-20-03398-f010:**
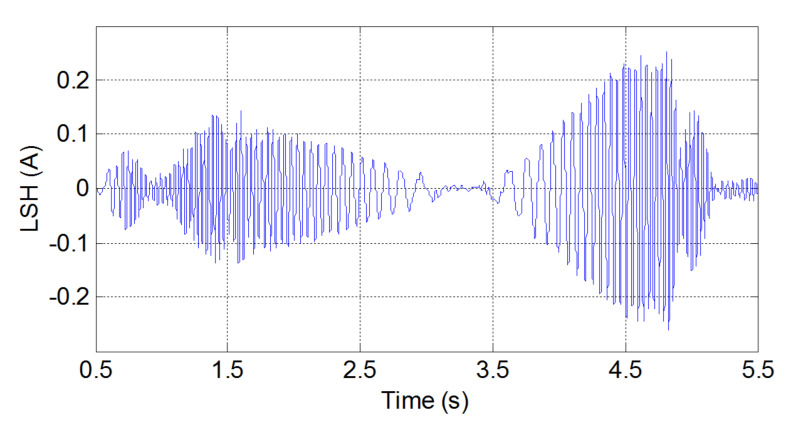
Experimental LSHst of the machine with one broken bar.

**Figure 11 sensors-20-03398-f011:**
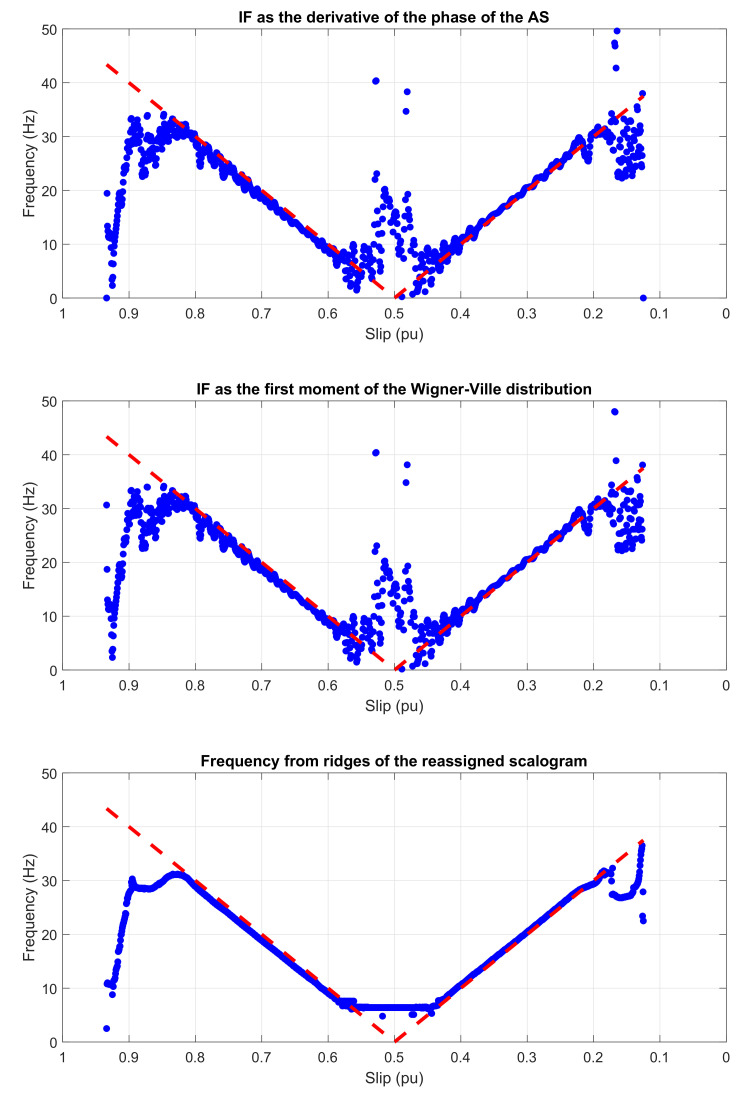
IF of the experimental LSHst computed using (from top to bottom): Hilbert transform, Wigner–Ville distribution and the reassigned scalogram based on the CWT. Dashed, red line: theoretical evolution of the fault harmonic. Dotted, blue line: evolution of the fault harmonic of the tested machine.

**Figure 12 sensors-20-03398-f012:**
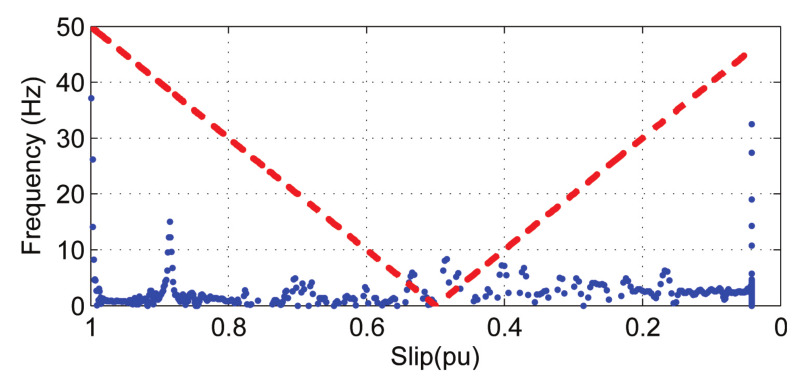
Experimental IF of the LSHst extracted from the current of a healthy machine. Dashed, red line: theoretical evolution of the fault harmonic. Blue dots: evolution of the fault harmonic of the experimental machine.

**Figure 13 sensors-20-03398-f013:**
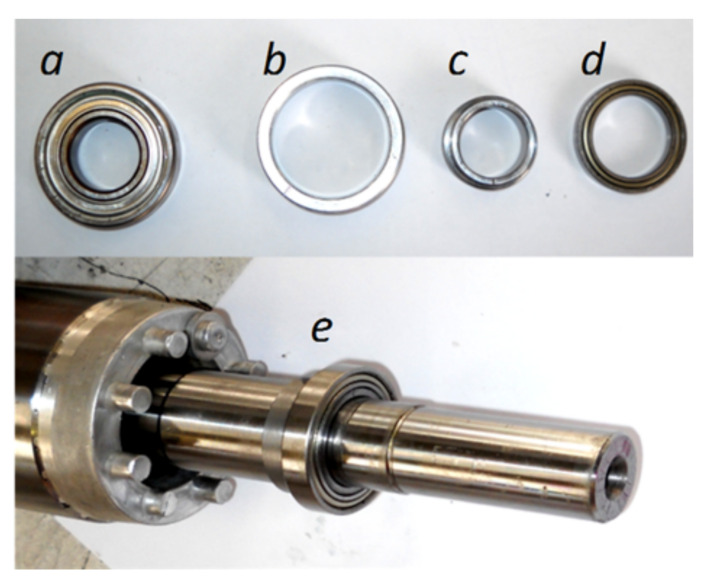
Rotor of the eccentric motor unit. Top, from left to right: (**a**) original bearing, (**b**) external and (**c**) internal eccentric rings, and (**d**) new bearing. Bottom: (**e**) mounted unit on the shaft.

**Figure 14 sensors-20-03398-f014:**
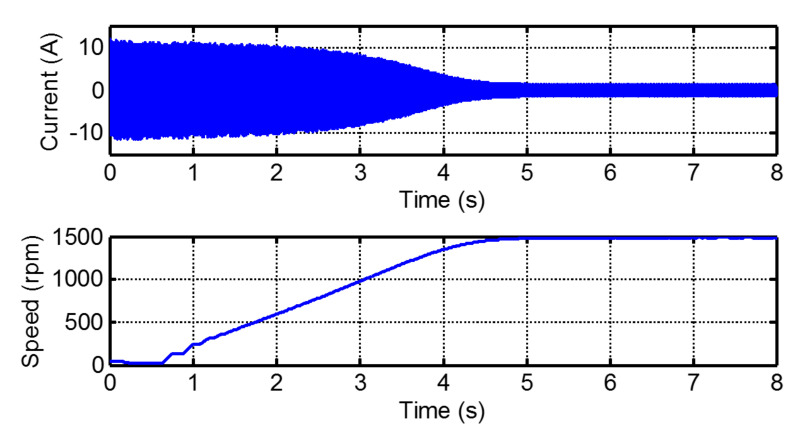
Current (**top**) and speed (**bottom**) measured in the experimental eccentric motor during the start-up transient.

**Figure 15 sensors-20-03398-f015:**
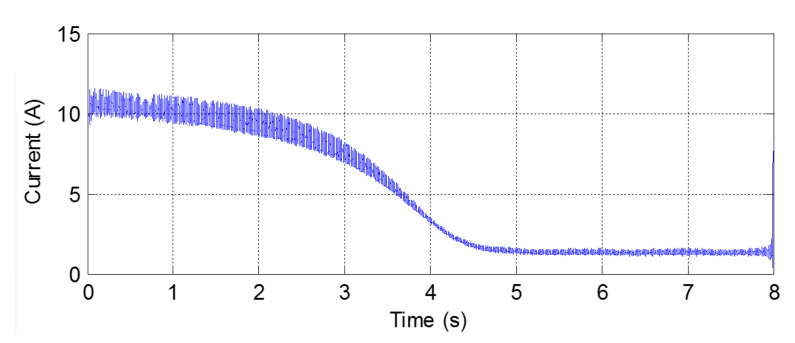
Envelope of start-up current extracted by means of HT.

**Figure 16 sensors-20-03398-f016:**
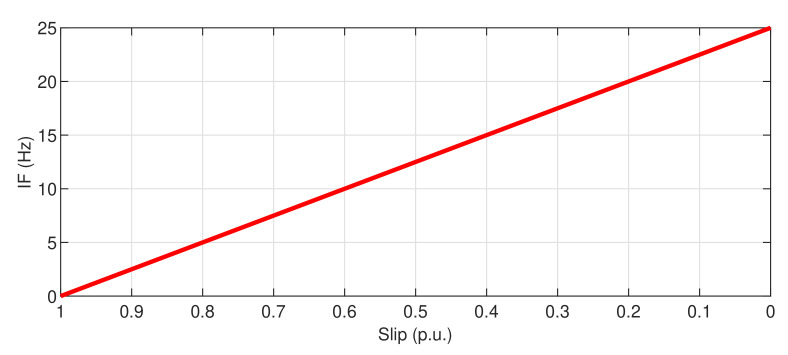
Theoretical evolution of the IF of the mixed eccentricity fault components in the envelope of the start-up current.

**Figure 17 sensors-20-03398-f017:**
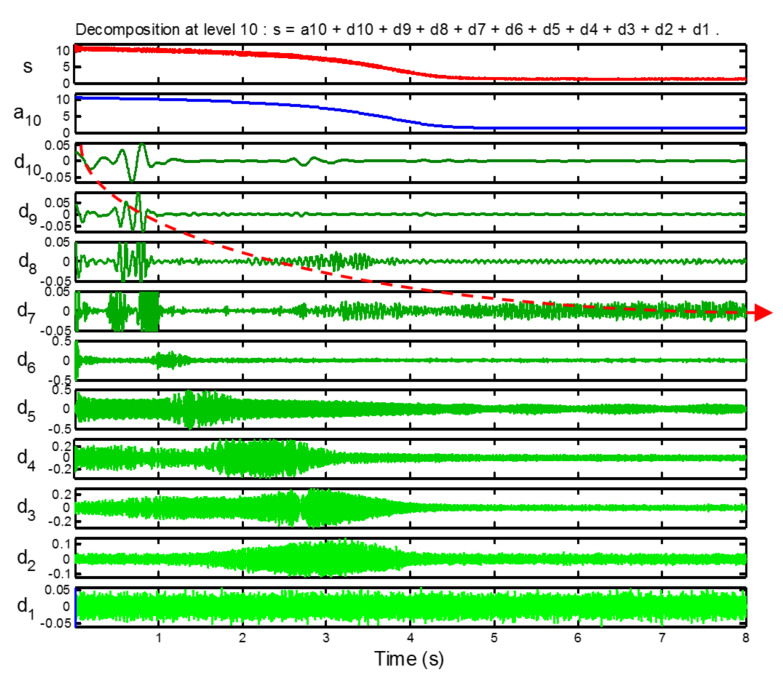
Wavelet decomposition of the start-up current; the components related to mixed eccentricity are included in the detail signals d7 to d10.

**Figure 18 sensors-20-03398-f018:**
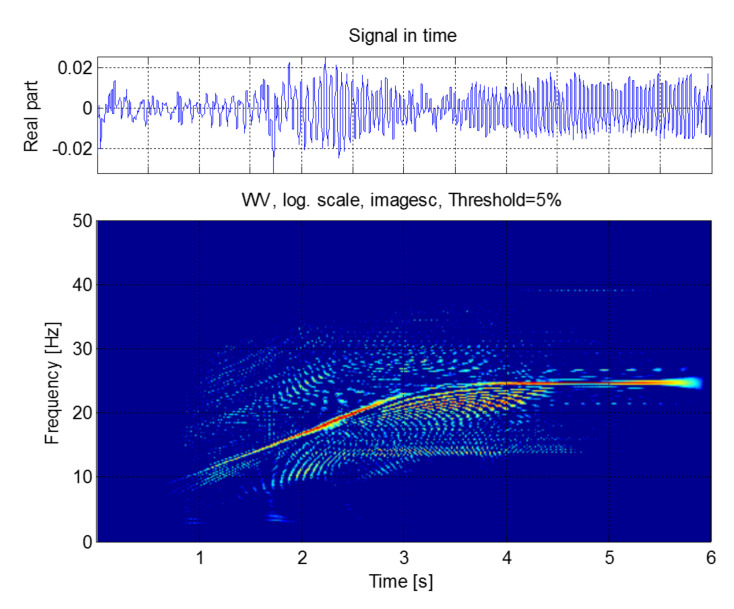
Components of the start-up current in the frequency band [2.5, 39.06] Hz, which includes the main mixed eccentricity components (**top**), and Wigner–Ville distribution of these components (**bottom**).

**Figure 19 sensors-20-03398-f019:**
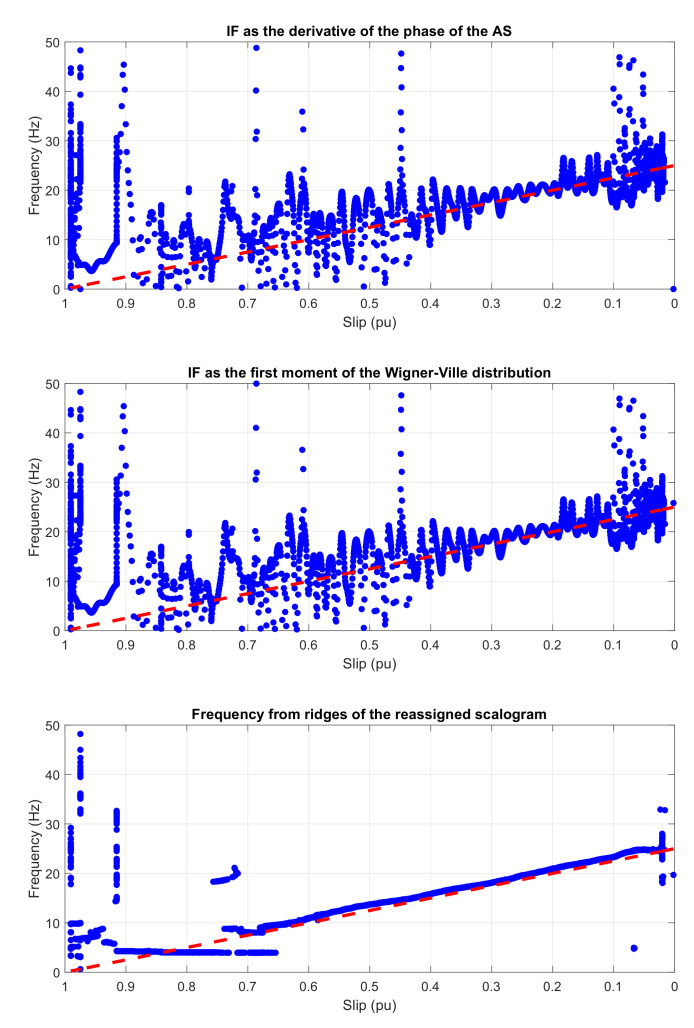
Instantaneous frequency versus slip of the main eccentricity components in a tested start-up calculated by means of (from top to bottom): Hilbert transform, Wigner–Ville distribution, and the scalogram based on the CWT. Dashed, red line: theoretical evolution of the fault harmonic. Blue dots: evolution of the fault harmonic of the experimental machine.

**Figure 20 sensors-20-03398-f020:**
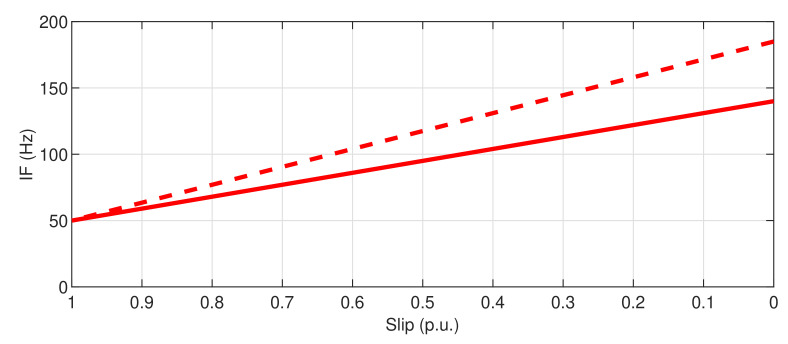
Theoretical signature in the slip–frequency plane of the IF of the bearing fault related harmonics, in case of a cyclic race fault in the outer race (solid line) and in the inner race (dashed line).

**Figure 21 sensors-20-03398-f021:**
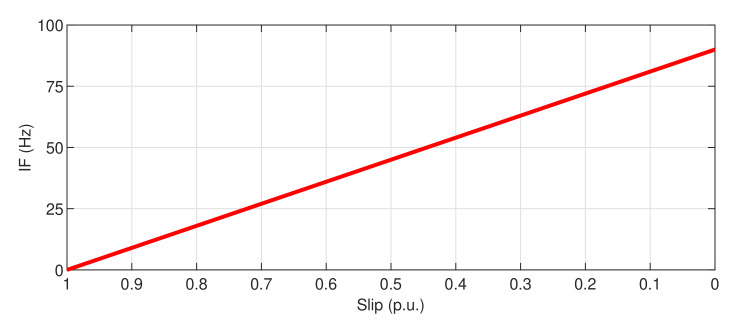
Theoretical evolution of the IF of the outer race fault components in the envelope of the start-up current.

**Figure 22 sensors-20-03398-f022:**
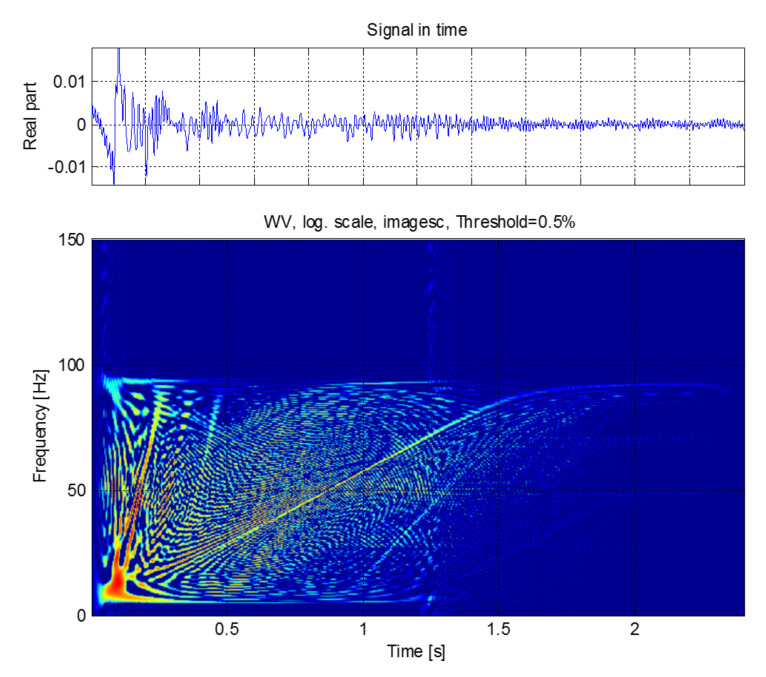
Components of the start-up current in the frequency band [2.93, 93.75] Hz, which includes the main outer race fault components (**top**) and Wigner–Ville distribution of these components (**bottom**).

**Figure 23 sensors-20-03398-f023:**
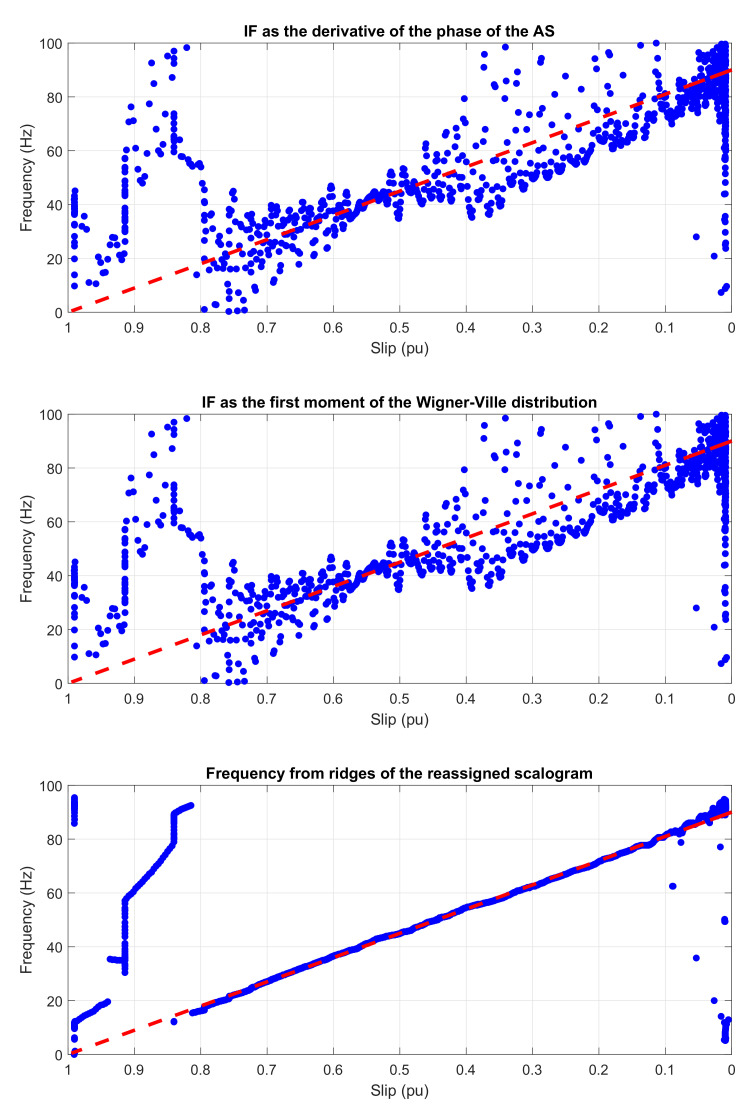
IF versus slip of the main outer race fault components of the start-up current calculated by means of, from top to bottom: Hilbert transform, Wigner Ville distribution, and the scalogram based on the CWT. Dashed, red line: theoretical evolution of the fault harmonic. Blue dots: evolution of the fault harmonic of the experimental machine.

**Figure 24 sensors-20-03398-f024:**
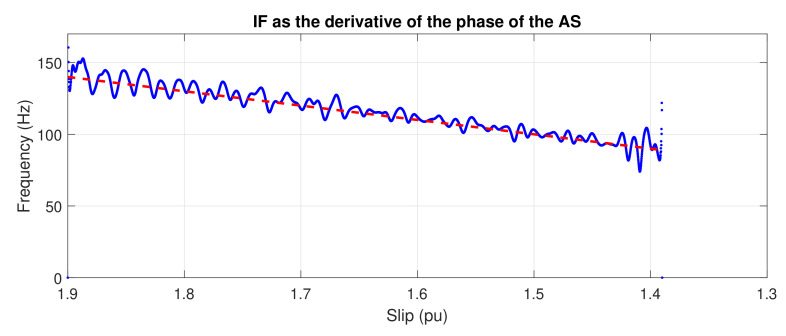
Instantaneous frequency of the LSH versus slip, under plugging conditions. Dashed, red line: theoretical evolution of the fault harmonic. Blue dots: evolution of the fault harmonic of the experimental machine.

**Table 1 sensors-20-03398-t001:** Comparison of the values of the correlation parameter $R_{\mathrm{IF}}$ computed with three different methods (HT, WVD, and CWT)

Machine	Fault Tested	HT	WVD	CWT
Faulty	Broken bar	0.97804	0.97804	0.9839
	(simulated)	(Figure 3)	(Figure 5)	(Figure 7)
	Broken bar	0.973	0.973	0.984
		(Figure 11 top)	(Figure 11 middle)	(Figure 11 bottom)
	Eccentricity	0.912	0.919	0.926
		(Figure 19 top)	(Figure 19 middle)	(Figure 19 bottom)
	Outer race	0.909	0.909	0.998
		(Figure 23 top)	(Figure 23 middle)	(Figure 23 bottom)
Healthy	Broken bar	0.007	0.007	0
		(Figure 12)		
	Eccentricity	0.147	0.143	0.009
	Outer race	0.050	0.046	0.097

**Table 2 sensors-20-03398-t002:** Computation time needed for calculating the IF

Method	Time (Relative to AS)
Analytic signal (AS)	1
Wigner–Ville Distribution (WVD)	18.4
Reassigned Scalogram	4023

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
