# Peer review of "Fault Diagnosis in the Slip–Frequency Plane of Induction Machines Working in Time-Varying Conditions"

_sensors, 2020, doi:10.3390/s20123398_

Round 1

Reviewer 1 Report

The manuscript proposes a method for fault diagnosis in the slip-frequency plane in induction machines in time-varying conditions. The proposed method would be useful for analyzing faults in a transient state. The manuscript also compares several methods with experiments. However, there are several concerns on

  1. Abstract should be revised to highlight key contribution of the manuscript.
  2. It is recommended to replace fist two figures in Fig.18 applying a low pass filter and delete Fig.19. There are too many figures which cannot provide much information in the manuscript.
  3. It is recommended to replace fist two figures in Fig.23 applying a low pass filter and delete Fig.24. It might enhance readability focusing on signal processing method proposed.
  4. Section 4 should include in-depth analysis and discussion for the proposed method. Based on the current form, it is hard to find key contribution.

Reviewer 2 Report

Topics related to non-invasive machine diagnostics through the use of non-destructive testing is important and current. Therefore, it is very important to introduce new non-invasive damage detection techniques in induction motors. However, some issues in the article should be clarified.
The article does not specify the accuracy (measurement uncertainty) with which the current supplied to the tested motor was measured. This is important because the tested machine was a 1.1 kW engine (short start-up time) and in the summary the authors write that the presented method is predisposed to higher power machines.
The font used in some Figures is too large - it is worth adapting it to the font size of the text in the article.
Is 0.4fr = 3.6fr in the formula (14) really mathematically correct?
It is good practice to refer to the Figures in the text before presenting the Figures.
Please correct grammatical errors in the article text.

Reviewer 3 Report

This paper has proposed a slip-frequency plane instead of the time-frequency plane for rotating machinery fault diagnosis. the paper is not easy to understand. Comments and suggestions are as follows:

  1. What is the physical meaning of 'slip' and how to obtain it?  It would be better to give more details about how to display the IF in the slip-frequency plane, otherwise, it is quite difficult to understand.
  2. About equations 3 and 4, why the fault frequencies of the bearing have a relationship with the power supply frequency f1? I read the references 21 and 22, the fault frequencies in [22] have nothing to do with f1. Please also check other equations.
  3. The bracket is in the sentence of 159 may be a typo.
  4. The name of the journal of the reference [54] is an error, please check it and also other references.

Reviewer 4 Report

Dear Authors,

Thank you to share this proposal. Please following these comments:

  1. Please add a block diagram in the introduction about your proposed algorithm and describe sub blocks briefly.
  2. The contribution of this manuscript is not clear, please explain it in more details in the introduction.
  3.  Why MCSA condition monitoring is used in this research? How about vibration /AE condition monitoring? Your technique can work with vibration or AE signals?
  4. How you can test the reliability and robustness of your proposed algorithm? it means: is your algorithm is robust? if yes how you can proof it?
  5. As the reviewer understood, the signal-based fault diagnosis is selected in the proposed method. The main issue of signal-based technique is uncertainty condition, so, how you can address this issue?

Good-Luck 

Round 2

Reviewer 3 Report

The paper has been revised according to my suggestions. The authors replied to my comments well. It can be published in this journal.

Reviewer 4 Report

Dear Authors,

Thank you for your cover letter. Regarding the second round review, this manuscript can be accepted to publish in this journal.

Best of Luck for your future research.